# Imprinting and Reproductive Health: A Toxicological Perspective

**DOI:** 10.3390/ijms242316559

**Published:** 2023-11-21

**Authors:** Ritu Chauhan, Anthony E. Archibong, Aramandla Ramesh

**Affiliations:** 1Department of Biochemistry, Cancer Biology, Neuroscience and Pharmacology, Meharry Medical College, Nashville, TN 37208, USA; rchauhan@mmc.edu; 2Department of Microbiology, Immunology and Physiology, Meharry Medical College, Nashville, TN 37208, USA; aarchibong@mmc.edu

**Keywords:** imprinting, epigenetics, endocrine disrupting chemicals, xenoestrogens, development, toxicants

## Abstract

This overview discusses the role of imprinting in the development of an organism, and how exposure to environmental chemicals during fetal development leads to the physiological and biochemical changes that can have adverse lifelong effects on the health of the offspring. There has been a recent upsurge in the use of chemical products in everyday life. These chemicals include industrial byproducts, pesticides, dietary supplements, and pharmaceutical products. They mimic the natural estrogens and bind to estradiol receptors. Consequently, they reduce the number of receptors available for ligand binding. This leads to a faulty signaling in the neuroendocrine system during the critical developmental process of ‘imprinting’. Imprinting causes structural and organizational differentiation in male and female reproductive organs, sexual behavior, bone mineral density, and the metabolism of exogenous and endogenous chemical substances. Several studies conducted on animal models and epidemiological studies provide profound evidence that altered imprinting causes various developmental and reproductive abnormalities and other diseases in humans. Altered metabolism can be measured by various endpoints such as the profile of cytochrome P-450 enzymes (CYP450’s), xenobiotic metabolite levels, and DNA adducts. The importance of imprinting in the potentiation or attenuation of toxic chemicals is discussed.

## 1. Introduction

The fetal origins hypothesis or the Developmental Origins of Health and Disease (DOHaD), also known as Barker’s hypothesis, suggests that environmental exposures during early life (particularly the in utero period) can permanently influence health and vulnerability to disease in in adulthood [1,2,3,4,5]. The extensive use of some of these chemicals that are structurally similar to estrogens for applications in medicine and agriculture [6] has resulted in an increased exposure of humans to endocrine-disrupting chemicals (EDCs) and, potentially, to their adverse health effects (e.g., infertility and cancer; [7,8,9,10,11,12,13,14]). Some of the synthetic estrogens and environmental chemical contaminants that possess estrogenic properties, also known as xenoestrogens, mimic the natural estrogens and bind to estrogen receptors (ERs) and, consequently, desensitize ERs or reduce the number of receptors available for binding. Thus, during embryogenesis, genetic factors that control the maturation of nerve endings in the hypothalamus are affected causing altered programming of sexual differentiation. The reproductive system is highly sensitive to the influence of excessive estrogen. Hence, the wide variety of xenoestrogens or EDCs may affect reproductive systems in adult animals and even cause reduced viability and fertility outcomes for the offspring [7,8,15,16,17,18,19,20,21,22]. The alterations that occur during critical stages of development are the result of long-term changes in the neuroendocrine system, referred to as ‘imprinting’ [23,24]. Imprinting occurs only after the effector has been metabolized and excreted [25]. The critical period of brain growth and development during which imprinting occurs is during the last trimester of pregnancy in humans [26,27], whereas in laboratory animal models, imprinting occurs during the last few days of pregnancy and the first week after birth [28]. During this vulnerable period, the fetus could potentially be exposed to high concentrations of imprinting agents such as drugs or hormone-mimicking toxicants in the environment [29]. The developmental window during which this imprinting occurs is brief, except in the case of the differentiation of genital organs [30]. Pre- and neonatal exposure to carcinogens and toxic environmental chemicals can increase the susceptibility of adult animals to biochemical insult in specific target organs [2,31,32,33,34]. Therefore, gaining a greater understanding of the mechanisms of imprinting and the effects of environmental chemicals on this process merits investigation.

## 2. Pathways

Most xenoestrogens that disrupt normal imprinting patterns act directly and indirectly. During the prenatal period, they directly interfere with organogenesis and inhibit the multiplication of primary germ cells. Consequently, the number of germ cells is significantly reduced and ultimately leads to improper morphogenetic differentiation. Some xenoestrogens act as endocrine-disrupting agents and interfere indirectly with hormonally mediated processes (hormonal imprinting [35]), such as the differentiation and development of genital organs and sexual behavior [36,37,38,39]. The mechanism is not fully understood and there is no clear demarcation between these two processes.

The neonatal period is associated with the sexual differentiation of organ systems [40]. Human male reproductive organs such as the testes are active during the first week of life, during which period androgenic imprinting occurs and plays a critical role in the development of male sexual characteristics [41,42,43]. Prenatal [44] and neonatal [45] exposure to toxic chemicals have been reported to affect the hypothalamic–pituitary–adrenal (HPA) axis [46,47]. The hormone-modulating pollutants cause organizational and activation effects, which render HPA to be sensitive to environmental chemical challenge in adulthood. Neonatal exposure to xenoestrogens affect reproductive hormone levels [48,49]. Reproductive hormones regulate the microsomal cytochrome P450 levels and vice versa [50,51]. Neonatal exposure to androgen-mimicking chemicals [52] results in imprinting by acting on P450 isozymes, which are developmentally induced and regulated by the pituitary growth hormone secretion [53]. Consequently, the aberrant hormonal signaling processes [54] result in decreased or increased toxicity [55]. Hence chemicals that interact with the testicular–hypothalamic–pituitary axis [56] not only control male reproductive functions but also the manifestation of toxicity when exposed to other chemicals.

This review focuses on physiological, biochemical, and molecular pathways altered in adulthood as a result of pre and neonatal exposure to xenoestrogens, and the role of these alterations in response to an exogenous chemical exposure in adulthood in both sexes. However, a greater emphasis is placed on the male sex in this review due to the following reasons. Spermatozoan is the only specialized cell that can deliver 23 paternal chromosomes to the oocyte and hence plays a critical role in embryo development [57]. Therefore, knowledge on toxicant-induced genome reprogramming during imprinting in male animals and the adverse reproductive outcomes in male and female offspring because of paternally derived factors constitutes the major focus of this review.

## 3. Effect of Imprinting on Sex Hormones and Sexual Behavior

Imprinting alters hormone levels, cell proliferation, and sexual behavior. A recent study investigated the effects of exposure to three common endocrine-disrupting chemicals, (bisphenol-A [BPA], 50 mg/kg bw/day; diethylhexyl phthalate [DEHP; 750 mg/kg bw/day], and dibutyl phthalate [DBP]), on the epigenetic and phenotypic changes in rats. The study found that exposure to these EDCs led to the altered imprinting of several genes related to metabolic, reproductive, and developmental processes, including genes involved in the regulation of male sexual behavior [58]. In another study, rats treated with allylestrenol during the fetal life (15th, 17th, and 19th day after conception) showed a profound increase in the sexual activity of adult male rats compared with controls. An increase in the number of copulatory individuals with multiple ejaculations were noticed in this case. On the other hand, the treatment of male rats with allylestrenol during the neonatal period (1st, 3rd, and 7th day after birth) resulted in a moderate decrease in sexual activity compared with controls. These rats showed an increase in the number of individuals with single ejaculation, and a frequency of mounting and intromission three times that of the controls [59]. The immoderate use of estrogens/steroids potentiate the receptors of the hypothalamus–pituitary–gonadal axis and modulate the sexual behavior of the male rats [60]. Environmental chemicals structurally similar to synthetic estrogens were found to elicit similar results in adolescent male rats when challenged under identical experimental regimens [59]. Hormone levels during fetal development affect sexual behavior in rats, which was exemplified in a study where male rats neonatally treated with allylestrenol showed high levels of serum testosterone relative to controls in adulthood [61]. On the contrary, mice neonatally treated with diethylstilbestrol had significantly lower serum testosterone levels and slightly lower 17β-estradiol levels in testicular homogenates compared with controls [62]. The discrepancy between these studies with regard to serum testosterone could be due to the type of synthetic estrogen and doses used and the duration of treatment. Pap and Csaba [61] used a single treatment of 17.5 µg allylestrenol/animal on day 1 whereas Ohta et al. [62] administered 5 µg DES for the first 5 days after birth. Further, when allylestrenol-treated rats were challenged with benzo(a)pyrene [B(a)P] in adult life, they failed to show any change in serum testosterone levels after 1 week but showed marked elevation in the levels of this steroid 3 weeks after treatment compared with controls, and allylestrenol only treated rats [63]. The authors attributed the increased testosterone levels long after challenging with B(a)P (3 weeks) due to the time taken for microsomal P450 to convert B(a)P to its reactive metabolites and hence a delay in influencing the hormone levels. While this explanation is plausible, without information on the time–course bioavailability of B(a)P, it is difficult to establish a cause–effect relationship.

One study [64] investigated the impact of neonatal exposure to BPA on the neural circuit responsible for regulating estrous cyclicity. Pups were subjected to injections of either a control or different doses of BPA (0.05 and 20 mg/kg/day) from postnatal day 1 to 7. When these female rats reached postnatal day (PND) 100, it was observed that 0.05 BPA mg/kg-exposed rats exhibited changes in their estrous cyclicity patterns, while 20 mg BPA/kg-exposed rats were unable to generate an estradiol-induced luteinizing hormone (LH) surge.

Bisphenol A exposure of rats during the neonatal period was reported to affect the hypothalamus. Real-time PCR analysis showed that the hypothalamic expression of mature LH-releasing hormone (LHRH) mRNA increased in the 0.05 mg BPA/kg/day group but decreased in the 20 mg BPA/kg/day group. Additionally, the study showed there was a reduction in unprocessed intron A-containing LHRH RNA in the cytoplasm of hypothalamic cells for both BPA 0.05 mg/kg/day- and BPA20 mg/kg/day-exposed rats. Further, the immunohistochemistry studies revealed an up-regulation of estrogen receptor alpha protein in the anteroventral periventricular nucleus and a down-regulation in the arcuate nucleus for both groups. These findings demonstrated that BPA permanently alters the processing of LHRH pre-mRNA and the estrogen-dependent mechanisms within the hypothalamus that control sexual behavior in adult female rats [64,65].

The exposure to sex hormones, such as testosterone and estrogen, during the neonatal period can also have a profound impact on the development of sexual characteristics and behaviors. In some species, exposure to specific hormones during this critical period can lead to sexual preferences and behaviors later in life [66]. A research group investigated the impact of neonatal androgenization, i.e., they administered 100 µg of testosterone propionate (TP) via subcutaneous injection to female mice on the day of their birth. The study focused on how this androgenization affected the growth hormone-releasing hormone (GHRH)–somatostatin–GH axis and its downstream targets, including liver enzymes and secreted proteins that are typically associated with either females or males. The findings suggested that exposure to steroids during the neonatal period contributes to changes in the GH axis and a reduction in the feminization of hepatic steroid-metabolizing enzymes, potentially affecting liver physiology [67]. An exposure to estrogen during the neonatal period has been found to have significant impacts on prostatic growth and increases the likelihood of developing prostatic intraepithelial neoplasia later in life [68]. 

## 4. Effect of Imprinting on Function and Structure of Reproductive Tissues

Aberrant imprinting can have significant effects on the structure and function of male reproductive tissues, leading to a range of reproductive disorders and conditions. Changes in imprinting can affect the regulation of genes involved in hormone production, which can lead to hormonal imbalances and further affect the structure and function of male reproductive tissues. This can result in conditions such as cryptorchidism, hypospadias, and testicular cancer. A study conducted on pregnant female Sprague–Dawley (SD) rats exposed to diethylstilbesterol (DES) on embryonic day 13.5, shows that it can inhibit transabdominal testicular descent in a dose-dependent manner. The insulin-like peptide 3 (INSL3; involved in Leydig cell differentiation in fetal testis) could have been responsible for testicular descent. The INSL3 may have been induced by down-regulating the expression of steroidogenic factor 1 (SF-1), a nuclear receptor required for gonadal development. HOXA10, a gene responsible for regulating morphogenesis and differentiation may not be associated with DES-induced intra-abdominal cryptorchidism at 2.5 mg/kg, but was associated with 5, 10, and 20 mg/kg. The human INSL3 receptor leucine-rich repeat-containing G protein-coupled receptor 8 (LGR8) could not be implicated in DES-induced transabdominal testicular descent [69].

Perinatal exposure to 17β-estradiol has been shown to cause lateral prostate inflammation in adulthood [70]. When pregnant mice were exposed to DES, the highest dose (100 mg/kg/day) showed reproductive tract abnormalities, including cryptorchidism on male offspring [71]. Similarly, in pregnant mice fed with 100 µg/kg body weight (bw) DES, there was an increase in prostate weight in adulthood. On the other hand, when fetal doses (doses that are above physiological range estrogenic activity and within a toxicological dose range) of DES were used, there was a reduction in adult prostate weight. It is emphasized that if toxic doses of xenoestrogens such as methoxychlor (20–2000 µg/kg bw), DES (0.02–2 µg/kg bw), and BPA (2–20 µg/kg bw) are used in testing for endocrine disruption, these high doses mask the endocrine disruption effects that would normally occur at physiologically relevant low doses [72]. Rats neonatally treated with estradiol benzoate (2 µg/kg bw) displayed poor reproductive capabilities (e.g., changes in sexual behavior and histopathologic abnormalities of reproductive organs [73]). Similarly, neonatal estradiol benzoate (5 µg/kg/pup) and DES (5 µg/kg/pup) treatment caused reduced prostate weight and induced pathological changes in the prostate. These changes included epithelial/stromal hyperplasia and a reduction in the diameters of glandular tubules and lumen. The changes were so irreversible that even after anti-estrogen treatment, they were not reversed [74].

In utero exposure to DES caused cryptorchidism (with the testes attached to the abdominal wall), semen abnormalities, testicular hypoplasia, and reduced fertility in human male offspring [75,76]. Fetal exposure to DES (100 µg/kg bw) also affected male sex differentiation. In male mice treated with DES, an incomplete regression of Mullerian ducts was observed [77]. In healthy unexposed animals, testes controlled the fetal regression of Mullerian ducts through the anti-Mullerian hormone. These authors attributed the incomplete regression to a diminished sensitivity of Mullerian ducts to the anti-Mullerian hormone. Prenatal DES treatment delayed the onset of Mullerian duct formation. This results in an asynchrony in the critical timing of Mullerian duct formation and subsequent regression.

Neonatal exposures also affected testes significantly. Rats treated with estradiol benzoate (0.5 mg/5 g bw and 1.0 mg/5 g bw) caused the dilation of rete testis, and a delay in the maturation of sperm [78]. Neonatal treatment with DES and ethinyl estradiol showed reduced testis weights, and the structural and functional development of the excurrent ducts in rats [79]. In hamsters neonatally treated with 17β-estradiol, testicular and accessory organ weights and serum testosterone levels mirrored those of adults. However, after normal pubertal development, all the neonatally DES-treated (100 µg/animal) hamsters exhibited multiple lesions in the reproductive tract in adulthood. Also observed were cryptorchidism, solid testicular tumors, multiple epididymal cysts, and the involution of accessory organs. Further, the disruption of spermatogenesis in the seminiferous tubules with no developed germ cells was noticed. The epithelial layer of the epididymis was involuted with a preponderance of multinucleated cells. Even the seminal vesicle morphology was abnormal. However, no change in circulating testosterone levels was observed [80]. 

The neonatal exposure of male rats to 2.4 µg BPA/pup was reported to cause aberrant methylation in insulin-like growth factor 2, a maternally expressed and paternally imprinted gene (Igf2-H19) imprinting region (ICR) in resorbed embryo compared with control viable embryos. Additionally, H19ICR hypomethylation was observed in the spermatozoa of these rats. This report is another example of epigenetic-related adverse effects where aberrant methylation in spermatozoa was inherited by embryo leading to post-implantation loss [81].

Studies were conducted at a genetic level to unravel the elements involved in altered functional aspects of male reproductive organs because of estrogen exposure. Neonatal DES treatment (2 µg/pup/day) increased the altered C-fos proto-oncogene expression in mouse prostate [82]. This gene is involved in growth and cellular differentiation. The expression of this gene in the posterior periurethral prostatic collecting ducts after estrogen treatment indicate that it is responsible for the morphological changes in mouse prostate. The neonatal estrogenization of male rat reduced transforming growth factor-β (TGF β) type I levels in the prostate epithelium [83]. Further, neonatal estrogenization prevented transient expression of p21, a cyclin-dependent kinase inhibitor that is inducible by TGF β in the prostate. These perturbations indicate a possible role for TGF β in the blockade of epithelial cell differentiation and the proliferation of periductal fibroblasts. The prenatal treatment of rats with DES (600 µg/kg bw) and octyl phenol (500 µg/kg bw) resulted in a reduction in the amount of P450 17-alpha hydroxylase, and steroidogenic factors in testes. On the other hand, the neonatal treatment of these rats with DES caused a reduction in the amount of follicle-stimulating hormone β subunit (FSHβ) and inhibin α subunit in pituitaries and testes, respectively. Further, a decrease in testis weights and seminiferous tubule diameters was observed [84]. DES exposure affects gene expression in testes early in life, mainly in the neonatal period. Also, DES affects the pituitary production of FSH resulting in the retarded maturation of seminiferous tubules and reduced Sertoli cell numbers.

Fiandanese et al. [85] conducted a study to investigate the impact of maternal exposure to the plasticizer di(2-ethylhexyl) phthalate (DEHP) and polychlorinated biphenyls (PCBs), both separately and together, on the reproductive function of male mouse offspring. Dams were exposed to 1 μg PCBs (101 + 118)/kg/day, 50 μg DEHP/kg/day, or the DEHP/PCB mixture in their diet throughout pregnancy and lactation. The results showed that the DEHP/PCB mixture had permanent effects on the reproductive health of adult male F1 mice, which were different from those observed with the individual compounds. The study revealed a complex interplay between the DEHP/PCB mixture and the male reproductive system, resulting in a unique set of effects, including synergy in altering the gross appearance and histological changes of the testis, and antagonism on the expression of genes involved in pituitary-gonadal cross-talk. On the other hand, there was a lack of interaction in sperm parameters and testosterone production [85]. 

The exposure of mice pups to 2 µg DES on post-natal days 1–5 and 20 µg estradiol led to tumor initiation, reduced sperm count, and infertility in adulthood [86]. Also, marked disruption in tumor suppressor (p53) and epigenetic regulator (NP95) transcripts were noted. Studies conducted with Wistar rats revealed that prenatal exposure to dexamethasone (0.2 mg/kg/day) contributed to the inheritance of inhibition in the testicular synthesis of testosterone [87]. Similar studies conducted with ICR mice revealed that prenatal exposure to propyl parabens (7.5-, 90- and 450 mg/kg/day) perturbed estrous cycle regularity, decreased serum estrogen and progesterone levels, and increased the number of atretic follicles resulting in ovarian aging [88].

## 5. Effect of Imprinting on Receptors and Bone Formation

Aside from sexual behavior, structural/organizational deformities, and functional irregularities of the reproductive tissues, bone mineral density, and mineral content are also controlled by sex hormones [89,90,91]. Neonatal exposure to allylestrenol (17.5 µg/animal) and DES (8.8 µg/animal) elevated the bone mineral content and bone density in male rats [92]. This finding suggests that osteoblasts and osteoclasts involved in bone formation and bone resorption, respectively, are influenced by the receptor binding, which is influenced by the neonatal estrogen treatment. A study on rats investigated the potential effect of dexamethasone exposure during the fetal stage on skeletal growth and bone mineral density in adult offspring. Pregnant rats were treated with 100 µg/kg dexamethasone on days 9, 11, and 13 of the gestation period. The male rats were found to have significant effects of dexamethasone treatment as compared with female rats and showed transient increases in crown-rump length and tibia and femur lengths at 3–6 weeks of age [93]. The prenatal [94] and neonatal [95] treatment of rats with allylestrenol (2 µg/animal) were reported to increase the number of glucocorticoid receptors in the thymus. This may have implications on the reproductive behaviors in exposed animals. 

## 6. Effect of Imprinting on Metabolism of Toxic Chemicals

Altered imprinting interferes with the growth hormone regulation of specific cytochrome P450 (CYP) isoforms. The CYPs are involved in the metabolism of toxic chemicals, and other xenobiotics. Male-specific CYP450 in adult animals are regulated by exposure to xenobiotics through the alteration of androgen concentrations during the neonatal period [96].

Studies were conducted by inducing imprinting in laboratory animals during this critical window of the neonatal period and investigating the profile of drug-metabolizing enzymes in these animals relative to controls. These studies were extended further by challenging the animals with toxic chemicals in adult life to find out whether the altered pattern of drug metabolizing enzymes is reflected in the pattern of metabolism of toxic chemicals as mentioned below.

Rats neonatally exposed to DES showed a decrease in the endogenous levels of some hepatic drug-metabolizing enzymes in male rats. DES (1.45 µmols)-treated rats showed a decrease in the endogenous levels of UDP-glucuronyl-transferases compared with controls. DES males, when treated in adulthood with 3-methylcholanthrene and phenobarbital, exhibited an altered metabolism in terms of the levels of enzyme activities and metabolite patterns [97]. These authors also found that DES males exposed to 3-methylcholanthrene in adulthood had higher aryl hydrocarbon hydroxylase activities and lower UDP-glucuronyltransferase activities. Similarly, rats treated with 1.45 µmols of DES on PND 2,4,6 and in adulthood with phenobarbital, 3-methylcholanthrene prior to DES treatment with 1 mg/kg aflatoxin B1 (AFB1) during adulthood revealed decreased AFB-DNA adducts. These studies revealed that a single DES treatment was able to influence the adduct formation and disposition. On the other hand, DES males exposed to phenobarbital had reduced CYP and glutathione transferase activities. Besides a reduction in CYP levels, DES males also showed a decrease in ethylmorphine N-demethylation, and ethoxyconmarin 0-deethylation activities, in addition to a significant inhibition of lipid peroxidation [98]. The long-term alterations in hepatic enzyme activities as a result of neonatal DES treatment is also reflected in DNA adduction levels when challenged with a toxic chemical in adulthood [99]. The DES males when exposed to 1 mg/kg bw AFB_1_ in adult life showed an increase in alpha-class glutathione transferases (alpha GST). Neonatal DES males showed a 35% reduction in AFB_1_-DNA adduct levels compared with controls. Neonatal DES treatment caused long term changes in detoxifying enzymes such as alpha GSTs that are responsible for increased detoxification of the reactive AFB_1_-8,9-epoxide preventing it from adducting with DNA [100]. Studies conducted in our laboratory also suggested a similar protective effect of neonatal DES (1.45 µmols) treatment against B(a)P administration in adulthood. When DES rats were challenged with B(a)P in adulthood, the quantitative distribution of B(a)P metabolites showed a specific pattern. The concentration of metabolites in organic fraction were low in liver, prostate, and testis, whereas the metabolite concentration in aqueous fraction were significantly increased in DES rats. This suggests that prior DES exposure prevented the formation of reactive metabolites (usually present in the organic fraction) and favored rapid detoxification (as seen in an increase in water-soluble metabolites; [101]). This trend was also reflected in the formation and persistence of B(a)P-DNA adducts. Benzo(a)pyrene-DNA adduct numbers were less in DES males relative to controls [101]. The findings of our laboratory, [101]), Zangar et al. [99], and Lamartiniere et al. [100] suggest that prior DES exposure modulates the metabolism of carcinogens such that DES provides a ‘protective effect’.

Additionally, neonatal DES exposure in male rats showed a differential response in CYP450 levels. The Western blot profiles revealed that in DES males the levels of CYP2C6 were increased 60% whereas in CYP3A2, they were decreased by 44%, relative to controls, but CYP2E1, CYP2B, and CYP2C13 did not show any changes [102]. Since CYP2C13 is a male-specific isoform, the absence of any change in its levels indicated that DES response is not under the influence of any male growth hormone secretion. This opens up the possibility that neonatal DES exposure may act through more than one regulatory mechanism [102]. Serum testosterone levels and testosterone metabolism were monitored in DES rats. No correlation was found between them [103]. Serum testosterone levels were not involved in the regulation of hepatic CYP450 enzymes. Protein kinase C (PKC) enzymes were reported to regulate CYP enzymes [104]. However, Zangar et al. [102] did not find PKC alpha levels altered in DES males. These findings further emphasize the hypothesis of Zangar et al. [102] that DES acts through more than one regulatory mechanism. 

## 7. Epigenetic Changes and Imprinting

In the post-genome era, there has been a significant focus in epigenetics study to better understand the hidden health risks associated with environmental toxicants [105,106] as epigenetic processes interfere with gametogenesis, fertilization, and embryonic development. Genomic imprinting involves epigenetic form of gene regulation, which operates through non-mutagenic pathways [6]. Genomic imprinting is implicated in disease development via methylation, and chromatin modification in a tissue-specific manner [107,108]. In addition to DNA methylation, histone modifications are also prone to disruption by xenoestrogens. Several key mechanisms such as methyl donor availability, loss of imprinting control, changes in dioxygenase activity, altered expression of noncoding RNAs, and the activation of cell signaling pathways that can alter the activity of histone methyltransferases have been proposed to explain the altered genomic programming, which can pass on to generations [109]. This “developmental reprogramming” induced by epigenetic changes during the pre- and neonatal period can have a lasting impact on an individual’s epigenetic regulatory aspects and health outcomes [110,111,112,113,114]. 

In one study, the molecular consequences of neonatal BPA exposure (2.4 μg per pup) in male rats (F0) was examined. The methylation status of the H19 imprinting control region (ICR) in resorbed embryos (F1) was examined and compared with the spermatozoa of their respective fathers (F0). The findings revealed a significant decrease in the expression of Igf2 and H19 genes in resorbed embryos (F1) associated with BPA exposure when compared with viable embryos in the control group. Additionally, significant hypomethylation at the H19 ICR in both the spermatozoa and resorbed embryos sired by rats exposed to neonatal BPA was observed. These results suggest that the abnormal methylation patterns at the ICR observed in spermatozoa were passed on to the embryo, leading to disruptions in the expression of Igf2 and H19 and ultimately contributing to post-implantation loss. This mechanism could be one of the potential ways through which BPA induces adverse epigenetic effects on male fertility [81]. Researchers have also studied the association between phthalates, heavy metals, and other EDCs’ exposure and DNA methylation patterns in various studies. While the results are not uniform across all studies, there is evidence suggesting a potential link between EDC exposure and altered methylation [115,116,117,118,119].

Pregnant rats treated with cypermethrin (5 mg/kg) from GD 5 to 21 and re-challenged with the same chemical (10 mg/kg for 6 days) revealed alterations in histone acetylation and DNA methylation in promoter regions of CYP1A and 2B-isozymes in brain tissues. The overexpression of CYPs may predispose the prenatal rat offspring to other toxicants exposed during adulthood and later [74].

The transplacental exposure of Sprague–Dawley rats to *p,p*′-DDE (a metabolite of DDT; 100 mg/kg bw) from GD 8 to GD 15 was reported to show abnormal testis histology and decreased sperm fertility indices with hypomethylation in H19 and Gtl2 genes in offspring, which was maintained in F1 and F2 generations [120].

The prenatal exposure of rat pups to lindane (0.25 mg/kg bw for 5 days), a hexachlorocyclohexane (HCH) isomer during GD 5–21 when rechallenged with lindane (5 mg/kg bw for 5 days) revealed elevated expressions of cerebral CYPs. Interestingly, alterations in histone H3 acetylation and DNA methylation associated with cerebral CYPs are indicative of the epigenetic regulation of CYPs, which is likely to enhance neurotoxicity in the offspring [121].

Urine and semen samples obtained from a cohort of human subjects and analyzed for hydroxylated metabolites of PAHs (1-hydroxy phenanthrene and 1-hydroxypyrene; biomarkers of PAH exposure) correlated well with some variant proteins in imprinted genes H19 and Meg3. These studies reveal that environmental toxicant exposure can methylate sperm DNA [122]. Further studies revealed that children fathered by men in the above-mentioned cohort, had low birth weights and H19 imprinting was implicated because this gene showed a negative correlation with other birth outcomes [123].

The levels of BPA detected in the first trimester of pregnancy in a Michigan mother–infant pairs cohort were linked to DNA methylation in cord blood which was found to be female-sex specific [124]. These studies revealed that the phenomenon of toxicant-induced epigenetic changes could result in disease development at a later stage in life. Similarly, in another cohort of Mexican–American mothers exposed to phthalates (revealed by urinary phthalate metabolite concentrations) resulted in DNA methylation of nine differentially methylated regions [116]. 

An examination of infant–mother pairs from the Newborn Epigenetics Study (NEST) Cohort in Durham County, NC, revealed a link between maternal blood concentration and CpG methylation of the MEG3 gene. This link had stronger associations registered in children born to African American women compared with Caucasian women or Hispanic women [125]. These studies strongly support the notion that prenatal Cd exposure results in aberrant methylation. Another research group also found similar associations between placental Cd concentrations and the placental expression of genes H19, NDN, CPA4, GRB10, ILK, and DLX5 with the effects being more prominent in female fetuses in the New Hampshire Birth Cohort Study (NHBCS) and indicates that imprinting causes developmental toxicity [126].

One more study using the Korean mother–child pairs revealed in utero exposure to organochlorine pesticides DDT and HCH led to imprinting-related aberrations in gene expression. The concentrations of *p,p*′-DDT (a metabolite of DDT) and β-HCH (HCH isomer) measured in maternal sera showed a positive association with the decreased methylation in GF2 and LINE-1 in the placenta, respectively [127].

Aside from organic chemicals, prenatal exposure to metals revealed the transgenerational inheritance of abnormal spermatogenesis. When F1 generation of pregnant CD-1 mice exposed to sodium arenite (85 ppm) were mated to untreated female mice, both the F1 and F3 generations displayed decreased sperm quality and histological abnormalities. The in utero exposure was found to affect the IGF-2 gene (responsible for the proliferation of cells and active during fetal development) and the H19 gene (maternally inherited and plays an important role in early development, altered through an epigenetic pathway) [128]. Another study reported that the sperm epigenome was sensitive to arsenic (As) exposure. Paternal non-occupational exposure to As was found to induce the altered methylation status of Meg3 gene (maternally expressed version) in human sperm DNA, which may have a bearing on pregnancy outcomes [129].

While the above-mentioned studies reported on the exposure of female mice to EDCs and the adverse effects that ensued in offspring, other studies reported that even pre-conceptional maternal exposure contributes to heritable changes in offspring. For example, the pre-conceptional exposure of mother to cyclophosphamide, a chemotherapeutic and immunosuppressive drug (10 µg/mouse IP on alternate days), for 15 days was reported to cause delayed growth and altered methylation of the following imprinted genes H19, IgF2r, and peg3 in oocytes of F1 and F2 mice [130].

Prenatal exposure to perfluoroalkyl substances (PFAS) resulted in DNA methylation changes in mesoderm-specific transcript (MEST)-imprinted genes in mother–infant pairs in the Taiwan Birth Panel Study. Higher PFAS exposures resulted in lower methylation levels and childbirth weight, with the effect being more prominent in girls than boys. These findings reiterate that prenatal exposure to PFAS affects the integrity of fetal growth through faulty imprinting [131].

Mice exposed to the insecticide chlordecone (100 µg/kg/day) E 6.5 to E 15.5 gave birth to offspring which showed delayed puberty and a decreased number of primordial and atretic follicles. In addition to aberrations in reproductive physiology, alterations in DNA damage-associated marker genes such as YH2AX, the transcriptional repressor gene ZFP57, and imprinting regulation gene TRIM 28 were also noticed. Additionally, the genes responsible for estrogen signaling and oocyte maturation were found to be downregulated in the ovaries of adult mice. This finding highlighted that altered epigenetic imprinting impairs the female reproductive functions [132]. Similar heritable DNA methylation of imprinted genes in the oocytes of mice offspring exposed to 0 to 40 µg of diethyl hexyl phthalate (DEHP; a plasticizer prevalent in household items and medical devices) during 0 to 18.5-days post coitum was noticed [133].

Exposure to some of the EDCs, which act as obesogens during the neonatal period, disrupt the male reproductive axis by interfering with metabolic pathways, which are key to spermatogenesis. These chemicals cause alterations, which become imprinted on the genes of male gametes and transmitted across generations, adversely impacting the male reproductive health [39]. Since epigenetic processes interfere with gametogenesis, it is reasonable to conceive that exposure to these toxicants during gametogenesis contributes to defects in spermatozoa [134]. Ancestral TCDD exposure in SD rats at 200 or 800 ng/kg bw during GD 8–14 resulted in the epigenetic transgenerational inheritance of the imprinted gene, the insulin-like growth factor in rat liver tissues. Additionally, alterations in the expression of DNA methyl transferases (DNMT1, DNMT3A, and DNMT3B) along with hepatic damage and the increased activities of serum hepatic enzymes in F1 and F3 generation, were recorded [135].

In utero exposure to EDCs during the “male programming window” mimic steroid hormone actions by disrupting testicular development and impacting reproductive health during puberty and adulthood either through genetic (direct) or epigenetic effects. The resultant effects include disorganization of prostatic epithelium, prostatic intraepithelial neoplasia, and testicular dysgenesis syndrome (hypospadias, cryptorchidism, and poor semen quality [136]). Male mice exposed to the combustion toxicant benzo(a)pyrene (BaP) 1.0 or 2.5 mg/kg twice a week for 12 weeks showed a reduction in methylation levels of the imprinting genes H19 and Meg3 in sperm DNA of F0, F1, and F2 generations of mice, indicating a transgenerational effect [137]. Another case of germline epigenetics was reported in mice. The intra uterine exposure of pregnant mice to 2,3’,4,4’,5-pentachlorobiphenyl (PCB118) resulted in morphological changes in seminiferous tubules and a higher rate of sperm deformity in male offspring mediated through altered methylation pattern of imprinted genes in the sperm [138].

Paternal exposure to brominated flame retardants such as polybrominated biphenyls (PBBs) may affect the health of their children. Exposure to PBB153 was shown to alter the epigenome in human spermatogenic cells by decreasing DNA methylation. The reduction in DNA methyl transferase activity leads to the faulty imprinting of genes and consequently, contributes to health issues in men exposed to PBB153 [139].

Polybrominated diphenyl ethers which were carried by micro and nano plastics were reported to alter sperm DNA methylation through interfering with the hypothalamic–pituitary–testicular axis, contributing to Leydig cell dysfunction and spermatogenesis [140]. Other toxicants, such as BPA (25 mg/kg bw), bis(2-ethylhexyl) phthalate (375 mg/kg bw), and dibutyl phthalate (33 mg/kg), in pregnant rats revealed the transgenerational inheritance of sperm quality decline (F1, F2, and F3 generations). The epigenetic analysis of offspring spermatozoa revealed the presence of differential DNA methylation regions, which supported the notion that the above-mentioned EDCs contribute to male infertility through altered DNA methylation mechanisms [141]. It is beyond the scope of this review to elaborate on the harmful effects of microplastics and nanoplastics on reproduction and development, which is discussed in detail in the reviews of Hong et al. [142] and Ullah et al. [143]. Interested readers could refer to these publications.

Studies conducted in our laboratory showed that the exposure of pregnant female rats between days 11 and 20 of pregnancy to inhaled BaP (75 µg BaP/m^3^) restricted intrauterine growth and fetal survival [144]. These actions of BaP were attributed to significant reductions in progesterone, estrogen, and prolactin influences on placental regulation of fetal growth resulting from altered genomic imprinting effects of the above-mentioned hormones on placental function [144,145].

In a separate study, we showed that adult female rats exposed to BaP via inhalation (100 µg BaP/m^3^), 4 h/day for 14 days, ovulated fewer ova versus unexposed controls (15.3 ± 2.0) when mated with age-matched vasectomized males. However, when a separate group of similarly exposed female rats were mated with age-matched intact males with proven fertility, they sustained a significant percentage of fetal loss compared with their unexposed control counterparts [146]. Interestingly, our group [147] demonstrated that a single oral dose of 5mg BaP/kg body weight resulted in the production of higher concentrations of reactive BaP metabolites in ovarian, compared with liver, tissues and we suggest that these metabolites induce the epigenetic dysregulation of genes responsible for follicle development and defects in ovarian function [148] in post BaP-exposed adult female rats. 

Our group also showed that sub-acute exposure to BaP (75 µg BaP/m^3^), 4 h/day for 10 days, induced indices of infertility by significantly reducing epididymal sperm progressive motility and density compared with controls without influencing testes weight [149]. However, when rats were subjected to sub-chronic (60 days) exposure to the above exposure concentration [150,151], indices of fertility were reduced in exposed versus control rats. Altered indices included reduced (1) testis wt.; (2) seminiferous tubular (ST) volume; (3) ST length; (4) total wt of interstitium/paired testis; (5) and intratesticular testosterone concentration and daily sperm production compared with controls. The reduced male fertility indices enumerated above may be associated with abnormal imprinting resulting from DNA methylation dysregulation [152] induced by BaP.

In conclusion, several factors play a role in altered imprinting. It includes the environmental agents involved, their doses, exposure stage (prior to conception, post conception, embryo development), exposure duration, single chemical or multiple chemicals involved, prior exposure history of an individual, role of genetic and epigenetic factors, etc. A schematic representation of the critical events during imprinting and the adverse consequences in adulthood are depicted in Figure 1.

## 8. Future Directions

In real life situations, an organism is subjected to chemical insult from conception to senescence. It is well known that a single chemical exposure during neonatal life can induce more than one CYP450 isoenzyme. The altered expression of a CYP450 isoenzyme may attenuate the effect of one toxic chemical and potentiate the effect of others. Some drug-metabolizing enzymes have a delayed expression as a result of exposures to xenobiotics at an early age. So far, laboratory studies have concentrated on the effect of a single xenoestrogen. Studies are warranted on the effect of mixtures on reproduction and development at a molecular level to unravel the sequelae of developmental modifications and the role of imprinting on these, subsequent to pre and neonatal exposure. In this regard, we can take advantage of multiomic (proteomics, transcriptomics, epigenomics, etc.) approaches to gain a greater mechanistic understanding of imprinting.

In the context of imprinting, the exposome concept offers exciting opportunities in the area of reproduction and development. The exposome represents the full spectrum of exposures in an individual’s life span [153,154]. It is well established that epigenetics play a major role in prenatal perturbations induced by toxic chemicals. Hence, some of the epigenetic processes, such as DNA methylation, could be used as a proxy to measure the early life exposome and its effects across the life course [155]. Since prenatal and neonatal periods, and adolescence, are vulnerable stages for the onslaught of toxicants on the reproductive system, and their effects are manifested much later in life, the exposomal imprint is key for the development of several diseases [156,157,158,159].

Whether genomic imprinting could be used for the clinical management of diseases is not clear. One could argue that information on the expression of genomically imprinted genes that are specific to parent-of-origin could be used for this purpose. This approach has some difficulties because specific sets of postnatal clinical signs overlap even though many of them are linked to aberrations at a distinct genetic site not to mention the false positive results. Additionally, based on genetics alone, the prediction of outcomes of a pregnancy is difficult [160]. In situations like this, in addition to genetic results, fetal imaging could be helpful to take care of minor clinical issues such as low pre- and postnatal growth [161]. These shortcomings notwithstanding, prenatal diagnosis could be of help in pregnancy management.

## Figures and Tables

**Figure 1 ijms-24-16559-f001:**
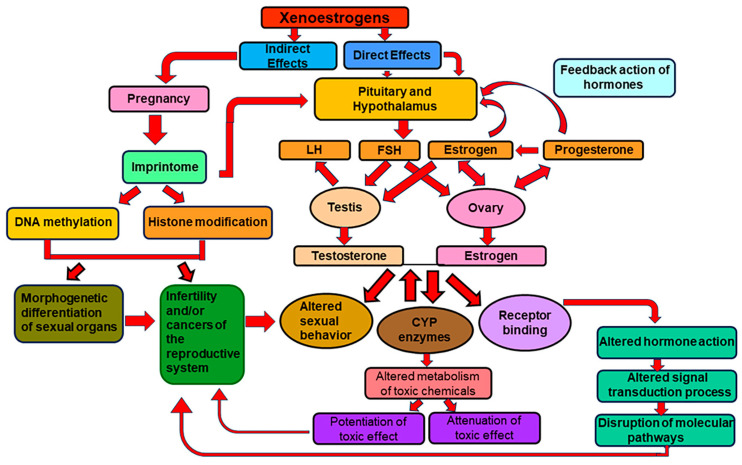
Schematic of the role of xenoestrogens on the developmental imprinting process and the consequences on reproductive system. Xenoestrogens or endocrine-disrupting chemicals contribute to imprinting through direct (genomic pathway) and indirect (epigenomic pathway) effects by acting on hypothalamo–hypophyseal axis. Since hypothalamus controls the secretion of reproductive hormones, faulty imprinting will result in aberrations in hormone-mediated pathways causing altered sexual behavior, infertility, and cancers of the reproductive system by triggering oncogenes and silencing tumor suppressor genes. Some of the hormones also influence the expression of cytochrome P450 (CYP) biotransformation enzymes, which, in a mixture exposure scenario, could alter the chemical metabolism causing either attenuation or potentiation of toxic effect. Some of the CYPs are also involved in the synthesis of hormones. The indirect effect of imprinting proceeds through the epigenetic pathway causing DNA methylation and histone modifications, further causing disruptions in the differentiation and functioning of sexual organs and culminating in infertility and/or cancer. LH: Luteinizing hormone; FSH: Follicle stimulating hormone; CYP: Cytochrome P450 enzymes.

## Data Availability

Not applicable.

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
