# Peer review of "Imprinting and Reproductive Health: A Toxicological Perspective"

_ijms, 2023, doi:10.3390/ijms242316559_

Round 1

Reviewer 1 Report

Comments and Suggestions for Authors

This review article provides a thorough analysis of developmental origins of human disease from a chemical exposure standpoint. It also provides detailed information on the implications of imprinting induced by environmental pollutants and other endocrine disruptors via genetic and epigenetic pathways. Overall, the manuscript is well-presented. However, I have the following suggestions/comments for enhancing the impact of this review in imprinting.

1.     Include the works of Fucic et al (2017) and Elbracht et al (2020) on imprinting in the “background” section. Please see below references.

Fucic A, Guszak V, Mantovani A. Transplacental exposure to environmental carcinogens: Association with childhood cancer risks and the role of modulating factors. Reprod Toxicol. 2017; 72: 182-190.

Elbracht M, Mackay D, Begemann M, Kagan KO, Eggermann T. Disturbed genomic imprinting and its relevance for human reproduction:n causes and clinical consequences. Hum Reprod Update. 2020; 26(2):1 97-213.

2.     Cite works of Yin et al (2022), and Lu et al (2023) on arsenic-induced imprinting and Ma et al (2019) on PAH-induced imprinting. Similarly, authors could incorporate findings of the research on Cd-, p,p’-DDE-, PBDE- and phthalate-induced imprinting.

Yin G, Xia L, Hou Y, Li Y, Cao D, Liu Y, Chen J, Liu J, Zhang L, Yang Q, Zhang Q, Tang N. Transgenerational male reproductive effect of prenatal arsenic exposure: abnormal spermatogenesis with Igf2/H19 epigenetic alteration in CD1 mouse. Int J Environ Health Res. 2022; 32(6): 1248-1260.

Lu Z, Zhao C, Yang J, Ma Y, Qiang M. Paternal exposure to arsenic and sperm DNA methylation of imprinting gene Meg3 in reproductive-aged men. Environ Geochem Health. 2023; 45(6): 3055-3068.

Ma Y, Lu Z, Wang L, Qiang M. Correlation of internal exposure levels of polycyclic aromatic hydrocarbons to methylation of imprinting genes of sperm DNA. Int J Environ Res Public Health. 2019; 16(14): 2606.

3. Mention findings from Agrahari et al (2019) on overexpression of CYP and its role in imprinting.

Agrahari A, Singh A, Srivastava A, Jha RR, Patel DK, Yadav S, Srivastava V, Parmar D. Overexpression of cerebral cytochrome P450s in prenatally exposed offspring modify the toxicity of lindane in rechallenged offspring. Toxicol Appl Pharmacol. 2019; 371: 20-37.

4. In the section on future directions for research, the authors are advised to include some information on if imprinting disorders could be corrected through prenatal testing.  Another aspect that is worthy of consideration is to discuss the role of exposome in the imprinting process.

Reviewer 2 Report

Comments and Suggestions for Authors

The manuscript, submitted for evaluation titled ‘Imprinting and Reproductive Health: A Toxicological Perspective describes the role of imprinting in the organism development and how exposure to environmental chemicals during fetal development leads to physiological and biochemical changes. The research area taken by the Authors is interesting but does not cover the latest issues. There are comments/questions that would be addressed to the Authors:

General Comments

The addition of information on disruption of female reproduction in the presence of xenoestrogens would be valuable. This is especially true for Figure 1. If the Authors were to focus on male disorders in the presence of EDCs, I would suggest changing the title, figure, and text.

The Authors focused on DES, BPA, PFAS etc. Their effects have been described many times and there are numerous reviews. The references do not include the most recent studies and reviews, which can be easily seen.

The information on the source of the endocrine disruptors mentioned in MS in the introduction could highlight the potential risk of exposure of the male organism to these substances and why it is worth addressing.

I would suggest mentioning these substances only in general terms, showing the mechanisms of action/toxicity in more detail, focusing on the latest studies, and describing the latest disruptors such as microplastics to make this review more up-to-date.

Specific Comments

Figure 1 -  the valuable graphic description of the subject. But information is missing or wrong:

a) References are missing or the illustration should be placed after the description of the subjects in the text.

b) I would suggest referring the changes to the pituitary gland in general – not just hypophyseal.

c) Large letters within the text in frames – please correct.

d) HPG axis – the arrows should lead to LH and FSH, not estrogen and progesterone. This is an important mistake. E2 and P4 should lead down to reproductive tissues. The Authors may include the feedback action of these hormones to the hypothalamus and pituitary.

(e) How does sexual behavior affect infertility and/or cancers of the reproductive organs? There is a mistake here. Please explain your position.

f) CYP enzymes also alter the synthesis of hormones – please add this.

In paragraph 3, titled “Effect of imprinting on male sexual behavior” the Authors describe not only sexual behavior. This paragraph should be rewritten and topics concerning physiology or the structure of the gonads should be included in the other paragraphs.

LL448-476 The data obtained in the previous studies are too detailed, and there is also a lack of a general conclusion summarizing the results obtained.

LL461-462;  474-476 Quite far conclusions.

Reference no. 36 and 37– missing data or mistake.

Author Response

Please see the atachment

Round 2

Reviewer 2 Report

Comments and Suggestions for Authors

The manuscript submitted for evaluation entitled ‘Imprinting and Reproductive Health: A Toxicological Perspective' contains all the necessary corrections and in my opinion could be accepted for publication.